# Groundwater Quality Assessment in the Northern Part of Changchun City, Northeast China, Using PIG and Two Improved PIG Methods

**DOI:** 10.3390/ijerph19159603

**Published:** 2022-08-04

**Authors:** Xinkang Wang, Changlai Xiao, Xiujuan Liang, Mingqian Li

**Affiliations:** 1Key Laboratory of Groundwater Resources and Environment, Ministry of Education, Jilin University, Changchun 130021, China; 2National-Local Joint Engineering Laboratory of In-Situ Conversion, Drilling and Exploitation Technology for Oil Shale, Changchun 130021, China; 3College of New Energy and Environment, Jilin University, Changchun 130021, China; 4Jilin Provincial Key Laboratory of Water Resources and Environment, Jilin University, Changchun 130021, China

**Keywords:** PIG, CRITIC-PIG, Entropy-PIG, groundwater quality assessment

## Abstract

As a numerical indicator, the pollution index of groundwater (PIG) has gained a great deal of popularity in quantifying groundwater quality for drinking purposes. However, its weight-determination procedure is rather subjective due to the absolute dependence on experts’ experience. To make the evaluation results more accurate and convincing, two improved PIG models (CRITIC-PIG and Entropy-PIG) that integrate subjective weights and objective weights were designed, and they were employed to appraise groundwater suitability for drinking purposes in the northern part of Changchun City. A total of 48 water samples (34 unconfined water samples and 14 confined water samples) with abundances of Ca^2+^ and HCO_3_^−^ were collected and tested to obtain the data for the analyses. The results showed that 60.4%, 47.9% and 60.4% of the water samples manifested insignificant pollution and were marginally potable based on the values of the PIG, CRITIC-PIG and Entropy-PIG, respectively. Though 48% of the water samples had different evaluation results, their level difference was mostly 1, which is relatively acceptable. The distribution maps of the three sets of PIG values demonstrated that the quality of groundwater was the best in Dehui City and the worst in Nongan County. Groundwater contamination in the study area was mainly caused by the high concentrations of TDS, TH, Fe^3+^, F^−^ and NO_3_^−^, which not only came from geogenic sources but also anthropogenic sources.

## 1. Introduction

Groundwater, as the premier and finite source of freshwater for human drinking purposes, is confronted with a more or less serious contamination status in many areas around the world on the grounds of the fast urbanization process and population growth, as well as the increase in anthropogenic activities (agricultural, industrial and domestic activities) [1,2,3,4,5]. Ample evidence has shown that a good correlation exists between the quality of groundwater and human health [6,7,8,9,10,11,12]. Therefore, the quantitative delineation of the regional groundwater contamination level for drinking purposes badly needs to be conducted for us to have an overall understanding of the water pollution status, which is helpful to carry out relevant and effective measures to maintain or improve water quality and ensure local inhabitants’ health.

The determination of each chosen parameter’s weight is an indispensable procedure in the process of the comprehensive evaluation of groundwater quality. The weight itself represents the importance of the evaluation parameter, and the larger the weight is, the greater the impact it has on the groundwater-quality-appraisal result. All in all, the subjective weighting method, the objective weighting method and the integrated weighting method are commonly used to determine the weight values in recent research studies [13]. Compared with the other two methods, the integrated weighting method overcomes the subjectivity caused by the complete dependence on experts’ subjective judgment as well as the illogicality caused by the total dependence on groundwater-quality-analysis data to some extent [14].

Initially proposed by N. Subba Rao [15], the pollution index of groundwater (PIG) is a useful and effective numerical indicator to quantify groundwater contamination for drinking purposes, and its application is widespread [16,17,18,19,20,21,22]. The subjective weighting method is employed in the process of determining the PIG value, and the calculation of the weight is based on subjective judgment and former experience. Several objective weight-determination methods, such as the Entropy-weighted model and the CRITIC (criteria importance through intercriteria correlation)-weighted model were integrated with TOPSIS and the WQI approach to evaluate water quality in different areas [23,24,25,26,27,28,29,30,31]. As for the integrated method, Zhang et al. integrated CRITIC (objective method) and AHP (subjective method) to calculate the index weight, aiming to comprehensively appraise the water-source vulnerability of Yuqiao Reservoir [32]. By integrating order relation analysis method and Entropy-weighted method, Gao et al. employed an additive model to evaluate the drinkability of groundwater in Xi’an city, Shaanxi Province [14]. Yan et al. improved the Entropy-weighting model by coupling the relative entropy theory to make the evaluation results more logical and reliable [33].

As a famous gold corn belt in China, from 70% to 80% of the population in the northern part of Changchun City (Dehui City, Yushu City and Nongan County) lives in rural areas, and groundwater is nearly the sole source for their drinking and irrigation aims [34]. Therefore, taking human health into consideration, it is worth carrying out groundwater quality assessment in the region to decide whether to take some measures to maintain or improve groundwater quality.

The present research study intends to: (1) propose two novel PIG models (the CRITIC-PIG model and the Entropy-PIG model) by integrating the traditional PIG with two objective weighting methods (the CRITIC method and the Entropy method, respectively); (2) employ the traditional PIG model and the two improved PIG models (the CRITIC-PIG model and the Entropy-PIG model) to evaluate the drinkability and obtain the overall pollution distribution of the groundwater in the northern part of Changchun City; (3) study the hydrochemical characteristics of groundwater in the study area by adopting graphical methods (Gibbs diagrams and Piper diagram). The result of the current study not only can provide the pollution status of groundwater for drinking purposes, which is helpful to carry out effective measures for the remediation and control of groundwater resources in Dehui City, Yushu City and Nongan County to guarantee the inhabitants’ health, but it can also offer two improved PIG models to assess the pollution levels of groundwater for drinking aims.

## 2. Overview of the Study Area

### 2.1. Study Area

Covering an area of 13,434 km^2^, the study area (124°32′–127°05′ E, 43°54′–45°15′ N) is located in the northern part of Changchun city (Nongan County, Dehui City, Yushu City), the hinterland of Songliao Plain, as illustrated in Figure 1. Songhua River, Yitong River, Yinma River, Mushi River and Kacha River are the five principal rivers flowing through. Denudation-accumulation high plain and accumulation-mountain-valley plain are the two major landforms of the study area, with elevation ranging from 130 to 296 m.

According to the Köppen climate classification, the study area lies in the temperate climate zone and belongs to the hot summer–dry winter temperate climate class (Dwa) with the temperature of the hottest month being over 22 °C. The annual average temperature, precipitation and evaporation from 1962 to 2000 were 4.3 °C, 538.6 mm and 1629 mm, respectively. Based on Figure 2, rainfall is mainly concentrated in the summer stage (from June to September), which accounts for approximately 78% of total annual rainfall, while evaporation mainly occurs from April to September, a period that accounts for around 80% of total annual evaporation.

### 2.2. Geology

The study area lies in the southeastern uplift area of the Songliao fault basin of Jihei Fold System, where the Lower Cretaceous and Quaternary strata are well developed, whereas most of the Cretaceous strata are covered by Quaternary loose deposits (sand, silt, subclay, gravel, pebbles). The lithology of the Cretaceous strata is dominated by mudstone, shale and silty sandstone of Nenjiang Formation (107–666 m), mudstone and sandstone of Yaojia Formation (35–220 m), mudstone and siltstone of Qingshankou Formation (32–196 m), clastic rock of Quantou Formation (29–2199 m) and mudstone and sandy conglomerate of Denglouku Formation (0–1082 m).

### 2.3. Hydrogeology

Groundwater in the study area is dominated by loose rock pore water (Quaternary) whose aquifer is mainly gravel, sand and loess, as well as clastic rock fissure–pore water (Cretaceous) whose aquifer is mainly sandy conglomerate, sandstone and mud rock [35]. The detailed hydrogeological information of each formation is listed in Table 1. Groundwater mainly receives recharge from the infiltration of atmospheric precipitation, and the major means of groundwater discharge are evaporation and artificial extraction for irrigation and drinking purposes.

## 3. Materials and Methods

### 3.1. Materials 

A sum of 48 wells (Figure 1), including 34 Quaternary unconfined water wells and 14 Quaternary confined water wells, were sampled in November 2017 to investigate the groundwater quality situation of the study area. To obtain accurate and reliable water-quality-analysis results, each well was extracted for 5–10 min before sampling to minimize the influence of residual water in the suction pipe. Samples were gathered in polyethylene plastic bottles (350 mL), which were pre-cleaned three times by using deionized water. Two bottles of water were gained from each well, and 10% nitric acid solution was added to one of them to make the pH less than two in order to perform a cation analysis, while the other one was non-acidified. pH and alkalinity were tested and determined in situ using the calibrated HANNA (HI99131) portable pH analyzer and Gran titration, respectively [36].

The water samples were carefully gathered, strictly sealed, clearly labeled and immediately transported to Pony Testing International Group in Changchun City to be tested. The water quality testing technique was in accordance with Chinese Drinking Water Standard Examination Methods (GB5750-2006). ICP-AES (inductively coupled plasma atomic emission spectrometry) and ion chromatography were used to examine the major cations (K^+^, Na^+^, Ca^2+^ and Mg^2+^) and anions (Cl^−^, NO_3_^−^, SO_4_^2−^, F^−^ and Fe^3+^), respectively. TDS and TH (Total Hardness; CaCO_3_ Hardness) were measured using the vapor-drying method (an electric blast-drying oven and an electronic analytical balance) and the Na_2_EDTA titrimetric method, respectively. All groundwater samples passed the reliability test (a charge-balance check), with the relative errors of the sum of anion and cation milliequivalent concentrations being less than 5%.

### 3.2. Methods

#### 3.2.1. The Traditional PIG Method

As is shown in Table 2, a total of twelve chemical parameters (TDS, TH, Ca^2+^, Mg^2+^, Na^+^, K^+^, HCO_3_^−^, Cl^−^, SO_4_^2−^, NO_3_^−^, F^−^, Fe^3+^) were chosen to carry out drinking-water quality appraisal in the study area. The procedures for computing the traditional PIG are summarized briefly in Figure 3 [15]. In Step 1, the allotted weight ($A_{w}$) (numbers from 1 to 5) was assigned by taking their respective importance for human health into consideration. The number itself quantitatively indicated the extent of impact on human health, so the larger the number was, the greater the impact it had. Step 2 calculated the subjective weight ($w_{sj}$) using the ratio of the allotted weight ($A_{w}$) of each parameter to the sum of all $A_{w}$ values. The ratio of the measured concentration (*C*) to the drinking-water standard ($D_{s}$) of its corresponding index was the result of the status of concentration ($S_{c}$). Overall water quality ($O_{w}$) for drinking purposes was gained via the multiplication of the subjective weight ($w_{sj}$) and the status of concentration ($S_{c}$) in Step 4. Then, in Step 5, after calculating the sum of $O_{w}$ for each water sample, their respective PIG was obtained.

#### 3.2.2. The Improved PIG Methods (the CRITIC-PIG Method and the Entropy-PIG Method)

Figure 4 shows the procedure of the determination of two distinct objective weights using the CRITIC [30] and Entropy [38,39] methods, respectively. $x_{ij}$, the element of Evaluation Matrix X, is the measured value of the jth parameter of the ith water sample. m(48) is the number of water samples, and n(12) is the number of parameters. $y_{ij}$ is the measured value after normalization, which ranges from 0 to 1. $\max(x_{ij})$ and $\min(x_{ij})$ refer to the maximum and the minimum of the jth parameter of the ith water sample, respectively. For the CRITIC method, information account $C_{j}$ is associated with $\delta_{j}$, the standard deviation of the jth parameter, as well as $r_{ij}$, the correlation coefficient of the ith and jth indicators. For the Entropy method, constant 0.0001 is used in the formula of $P_{ij}$, aiming to avoid meaninglessness when $y_{ij}$ is zero.

As is shown in Figure 5, the improved PIG methods (the CRITIC-PIG method and the Entropy-PIG method) were adopted to integrate subjective weights (listed in Table 2) and objective weights (CRITIC or Entropy method), which not only involve human subjective judgment but also objective calculation in Steps 1 and Step 2 [40], while the other procedures are essentially the same as the traditional PIG method. After calculation, the CRITIC-PIG value and the Entropy-PIG value were obtained. Based on the values of three PIG values, they were divided into five categories (Table 3).

## 4. Results and Discussion

### 4.1. Physicochemical Parameter

Based on the statistical analysis data (Table 4) of the main ion concentrations as well as the major water quality indexes of the 48 groundwater samples (confined water and unconfined water) in Dehui, Nongan and Yushu Districts, unconfined water and confined water manifested a weak alkaline environment, with pH ranging from 6.7 to 8.5 and from 7 to 7.8, respectively. The pH value was relatively stable due to the low S.D. value (0.4 and 0.3, respectively). On the whole, unconfined water in the study area was characterized by higher concentrations of Ca^2+^, Mg^2+^, Na^+^, K^+^, Cl^−^, SO_4_^2−^ and HCO_3_^−^ than those in confined water. The box diagram of the eight major ions (except K^+^) of the two distinct types of water shown in Figure 6 indicates their analogous abundance order: cations dominated by Ca^2+^, followed by Na^+^ and Mg^2+^; anions dominated by HCO_3_^−^, followed by Cl^−^, SO_4_^2−^ and NO_3_^−^.

### 4.2. Spatial-Distribution Characteristics of TDS, TH, NO_3_^−^-N, Fe^3+^ and F^−^ in the Study Area

Figure 7 vividly presents the spatial distributions of TDS, TH, NO_3_^−^, Fe^3+^ and F^−^ in the area of consideration. TDS varied from 182 to 2280 mg/L, and Nongan County tended to have a higher concentration of TDS than Yushu City and Dehui City. Areas with high TDS were mainly located in the south and northeast of Nongan County. The spatial distributions of TH and NO_3_^−^ were similar to that of TDS, varying from 90.8 to 1200 mg/L and from 0.01 to 143 mg/L, respectively. As for Fe^3+^, a majority of areas had low Fe^3+^ content, except for the northeastern parts of Nongan Country and Yushu City. The presence of high Fe concentrations is closely associated with the Fe-rich matters in the reducing environment of the aquifer, the formation of organic complexes, which lead to the dissolution of Fe, and poor groundwater run-off conditions; however, the influence of pH on Fe in the study area is negligible according to Oluwafei Adeyeye’s quantitative analysis [41], which can explain the existence of elevated values of Fe under neutral–alkaline pH conditions. The content of F^−^ was relatively low in Yushu City and Dehui City, but Nongan country had the tendency of having high levels of F^−^, especially in the middle part.

### 4.3. Graphical Methods

Proposed by Piper [42], the Piper diagram is a commonly used tool to show the main hydrochemical types of a large number of groundwater samples from a specific area. As is shown in Figure 8, three major cation and anion types as well as a mixed type (non-dominant type) are listed as 1–7 in the bottom two triangles. Five categories of hydrochemical types are labeled A–E in the middle diamond. The bottom-left triangle indicates that the dominant cation in the vast majority of samples was Ca^2+^, and in the minority of samples, this was Na^+^ or a mixed type (non-dominant cations). The bottom-right triangle indicates that the dominant anion in the vast majority of samples was HCO_3_^−^, while some had no dominant anions (mixed type), and three water samples abounded in Cl^−^. Based on the diamond in the middle, the hydrochemical type of groundwater in the study area was relatively diverse as a whole, with the dominance of the HCO_3_-Ca type, as well as some Cl-Ca type, mixed type and a few HCO_3_-Na type.

Gibbs diagrams were initially proposed by Gibbs [43] to study the hydrochemical evolution characteristics of surface water, and their usage is extended to the field of groundwater studies nowadays [44,45,46]. Though this widely used and mainstream method remains controversial in the interpretation of groundwater chemistry [47], it roughly provides the overall tendency of the evolution of groundwater chemistry when combined with the Piper diagram, as in this study. Based on the relationships between TDS and Na^+^/(Na^+^ + Ca^2+^), and TDS and Cl^−^/(Cl^−^ + HCO_3_^−^), respectively, three genres of mechanisms affecting the chemical composition of natural water could be determined: precipitation dominance, rock (lithology) dominance and evaporation dominance [43]. According to Figure 9, a great quantity of groundwater samples were in the “rock dominance” section, indicating that water–rock interaction (rock weathering and leaching) was the major factor controlling the chemical types of groundwater. In addition, a small amount of unconfined water samples had a tendency towards evaporation dominance with the characteristics of high Cl^−^ and TDS levels. This is mainly caused by the arid and semi-arid climate with little precipitation and relatively intensive evaporation effects.

### 4.4. Results of PIG, CRITIC-PIG and Entropy-PIG

According to 12 groundwater quality indexes (TDS, TH, Ca^2+^, Mg^2+^, Na^+^, K^+^, HCO_3_^−^, Cl^−^, SO_4_^2−^, NO_3_^−^, F^−^ and Fe^3+^) and their respective drinking-water standards listed in Table 2, the groundwater quality appraisal for drinking purposes was completed using the PIG model and the two improved PIG models (CRITIC-PIG and Entropy-PIG). The results are listed in Table 5 and Table 6, including 34 unconfined water samples and 14 confined water samples, respectively. The PIG values ranged between 0.204 and 7.114, with an average of 1.162, and based on the traditional PIG model, using which those 48 samples could be classified into five categories, among them, 60.4%, 18.8%, 8.3%, 6.25% and 6.25% showed insignificant, low, moderate, high and very high pollution, respectively. As for the two improved PIG models, the CRITIC-PIG values were between 0.294 and 2.795, and the average was 1.216. It was calculated that 47.9%, 20.8%, 18.8%, 6.25% and 6.25% of them indicated insignificant, low, moderate, high and very high pollution, respectively, with respect to the results. The results of the Entropy-PIG model indicated that the minimum and maximum of the Entropy-PIG values were 0.229 and 3.985, respectively, with an average of 1.081. The percentages of water samples showing insignificant, low, moderate, high and very high pollution were 60.4%, 18.8%, 10.4%, 8.3% and 2.1%, respectively.

Obviously, the confined water samples showed better quality for human drinking than the unconfined ones as a whole, as can be seen by comparing Table 5 and Table 6, which can be proved by the percentage of the “Insignificant Pollution” ones (overall, 92.8%, 85.71% and 100% of confined water samples manifested insignificant-pollution status, which is suitable for human drinking, using the traditional PIG model, the CRITIC-PIG model and the Entropy-PIG model, respectively, while these values were 33.3%, 22.9% and 31.3% for unconfined water).

By applying those three models, the classification results were not totally consistent with each other, with 25 of 48 water samples having the same evaluation results. Considering their respective consistency, between the PIG model and the two improved PIG models, the consistency values were 56.3% (CRITIC-PIG) and 79.2% (Entropy-PIG) and between the two improved models, this was 62.5%. As for those samples having divergent evaluation results, the level difference was mostly 1, which demonstrated the relatively convincing and correct results.

### 4.5. Distribution Map of Three PIG Values

The spatial-distribution maps of PIG, CRITIC-PIG and Entropy-PIG are plotted in Figure 10. Compared with Yushu City and Nongan County, the groundwater pollution level in Dehui City was relatively low, with the predominance of insignificantly polluted areas and lowly polluted areas based on the three models. Yushu City showed a progressive increase in the pollution level from the southwestern part to the northeastern part by and large, while this tendency was not obvious in the distribution map of the CRITIC-PIG values. Combined with Figure 7, it was concluded that the high level of pollution resulted from the high concentration of Fe^3+^ in the northeast. As for Nongan County, lowly and moderately contaminated regions occupied a large proportion, and highly and very highly polluted areas were spread in the northeastern and southern parts, which was due to the high levels of TDS, TH, NO_3_^−^ and F^−^ contents, as can be seen in Figure 7.

### 4.6. Sources of Pollution

Judging whether the O_w_ (overall water quality) value is over 0.1 provides a means to determine the general source of pollution [15,48]. If the value is below 0.1, pollution mainly comes from geogenic sources, while if the value is over 0.1, pollution caused by anthropogenic activities could not be negligible. Considering the five pollution-level zones, Table 7, Table 8 and Table 9 list the average O_w_ values of the 12 chemical indexes obtained using the PIG, the CRITIC-PIG and the Entropy-PIG models. The results showed that the O_w_ value of the 12 chemical indexes generally tended to increase from the insignificant-pollution level to the moderate-pollution level, while for the high and very high levels, this tendency was not noticeable because the total number was relatively limited and they had some very high values of certain indexes. It was concluded that the higher the pollution level was, the greater impact human activities had on the deterioration of groundwater quality. From the analyses using the PIG, CRITIC-PIG and Entropy-PIG models, 87.5%, 97.9% and 91.7% of the water samples, respectively, had at least one parameter’s O_w_ value over 0.1, indicating that the pollution contribution in the study area was not only ascribable to geogenic sources but also to anthropogenic sources. Considering the results of the Gibbs diagrams, geogenic sources were mainly the weathering and dissolution of rocks and minerals. Based on the background of the study area, anthropogenic sources were mainly agricultural activities such as the excessive use of chemical fertilizers and pesticides, industrial activities and domestic waste. As a well-known corn belt zone, the intense use of agrochemical products, especially phosphate and nitrogen fertilizers, in the study area significantly elevates the concentrations of NO_3_^−^, which can be seen from Figure 7, and is likely to be associated with the occurrence of potentially toxic elements in groundwater (e.g., As, Cd, Cr, Cu, Zn), according to cutting-edge research [49], which thus degrade groundwater quality and affect human health.

## 5. Conclusions

Considering the subjectivity of the traditional PIG values, the two improved PIG methods, which combine the subjective weight and the objective weight, were utilized to determine the groundwater suitability for drinking purposes in the northern part of Changchun City. In addition, graphical methods (Piper diagram and Gibbs diagrams) were employed to study the hydrochemical characteristics of groundwater. The major conclusion are listed below.

(1)Showing to be weakly alkaline, groundwater in the study area abounded in the HCO_3_-Ca type. According to the Gibbs diagrams, the chemical composition of groundwater was dominated by water–rock interaction, with a small fraction of water samples being controlled by evaporation processes.(2)The values of the PIG, CRITIC-PIG and Entropy-PIG ranged from 0.204 to 7.114, from 0.294 to 2.795 and from 0.229 to 3.985, respectively, and classified 60.4%, 47.9% and 60.4% of the water samples into insignificant pollution; 18.8%, 20.8% and 18.8% of the water samples into low pollution; 8.3%, 18.8% and 10.4% of the water samples into moderate pollution; 6.25%, 6.25% and 8.3% of the water samples into high pollution; and 6.25%, 6.25% and 2.1% of the water samples into very high pollution. In total, 52% of the water samples had the same evaluation results based on the three methods, and the same evaluation results occurred in the percentages of 56.3%, 79.2% and 62.5% between PIG and each of the two methods, and between CRITIC-PIG and Entropy-PIG, respectively. The level difference among the samples having different results using the three models was mostly one, which indicated that the results were relatively convincing.(3)Pollution came not only from geogenic sources (weathering and dissolution of rocks and minerals, evaporation) but also anthropogenic sources (agricultural activities, industrial activities and domestic waste) based on the O_w_ (overall water quality) index.(4)The distribution map of the three PIG values demonstrated that groundwater in Dehui City was the most suitable for drinking, with the dominance of insignificantly and lowly contaminated regions. Yushu City showed a progressive increase in the pollution level from the southwestern part to the northeast by and large, and the high-pollution areas were mainly affected by the high concentrations of Fe^3+^ in the northeast. Occupying a large area of lowly and moderately polluted regions, groundwater quality in Nongan County was worse than that in the other two cities. High levels of TDS, TH, NO_3_^−^ and F^−^ contributed to highly and very highly polluted groundwater in the northeastern and southern parts.(5)The results of the present research study provided an overall groundwater pollution status for drinking purposes in the north of Changchun City, which could be useful for the relevant authorities to take some protective and remedial measures for the guarantee of high-quality drinking groundwater for the people. However, due to the lack of sufficient water samples, a further groundwater quality investigation needs to be carried out in the study area, especially in those places whose PIG, CRITIC-PIG or Entropy-PIG values were over 1, aiming to obtain more accurate results.

## Figures and Tables

**Figure 1 ijerph-19-09603-f001:**
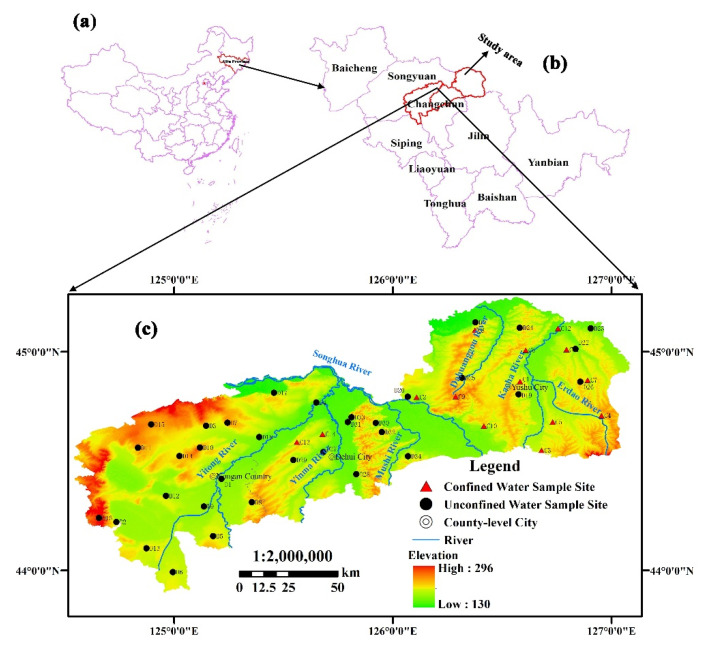
Location of the study area in the northern part of Changchun City (Nongan County, Dehui City, Yushu City) in Jilin Province, China. (**a**) Jilin Province, China; (**b**) the location of the study area in Jilin Province; (**c**) the terrain and sampling point distribution map of the study area.

**Figure 2 ijerph-19-09603-f002:**
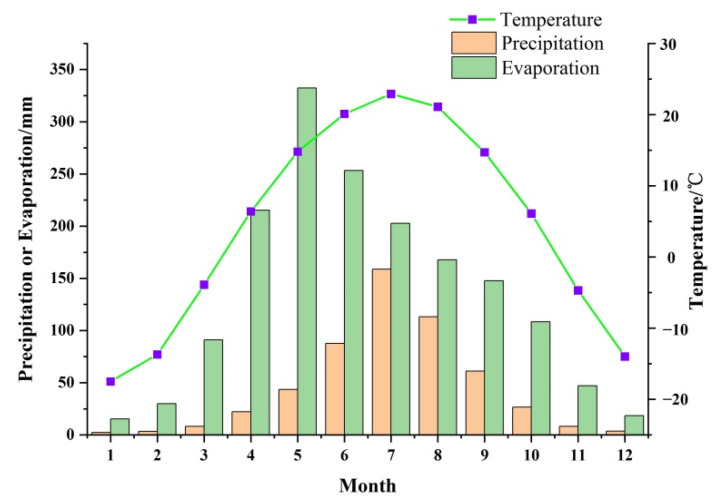
Monthly average values of precipitation, evaporation and temperature in the northern part of Changchun city from 1962 to 2000.

**Figure 3 ijerph-19-09603-f003:**
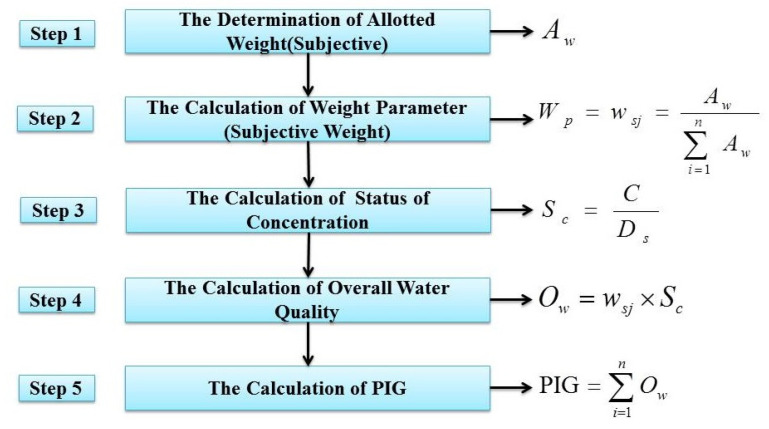
Technical roadmap of the traditional PIG method.

**Figure 4 ijerph-19-09603-f004:**
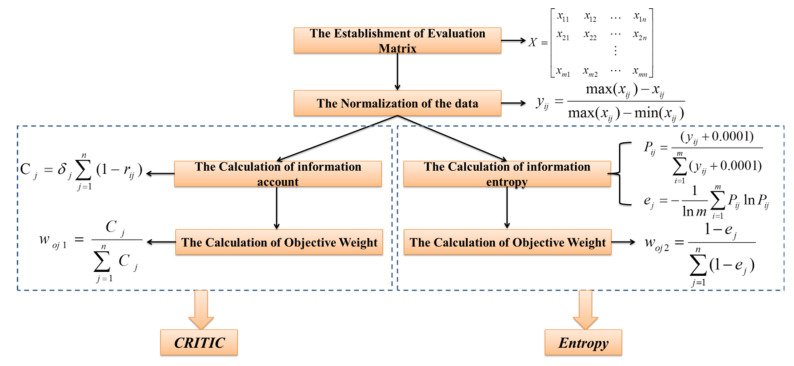
Technical roadmap of objective weight determination using the CRITIC and Entropy methods, respectively.

**Figure 5 ijerph-19-09603-f005:**
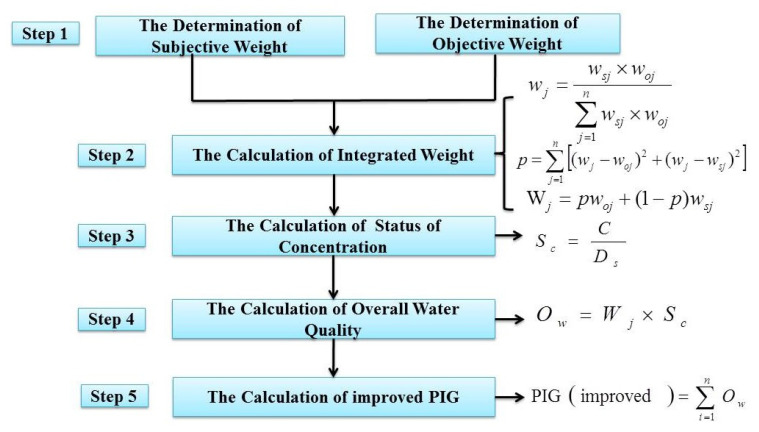
Technical roadmap of the improved PIG method.

**Figure 6 ijerph-19-09603-f006:**
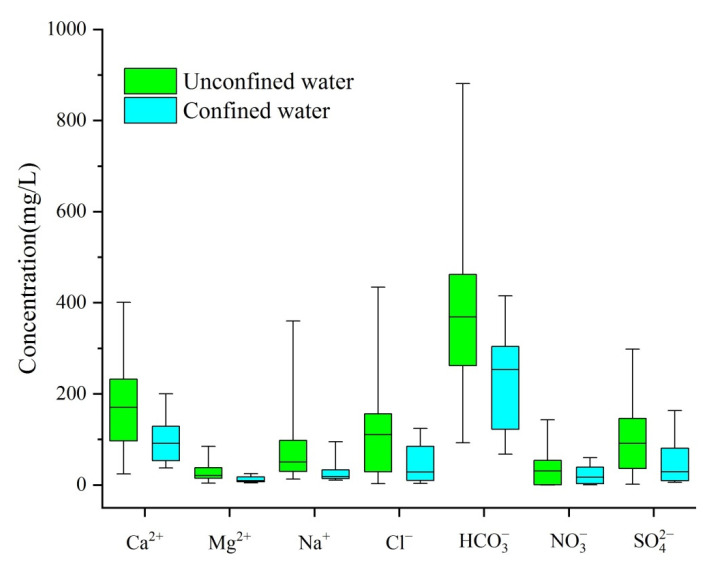
Box diagram of 7 main ion concentrations in confined water and unconfined water.

**Figure 7 ijerph-19-09603-f007:**
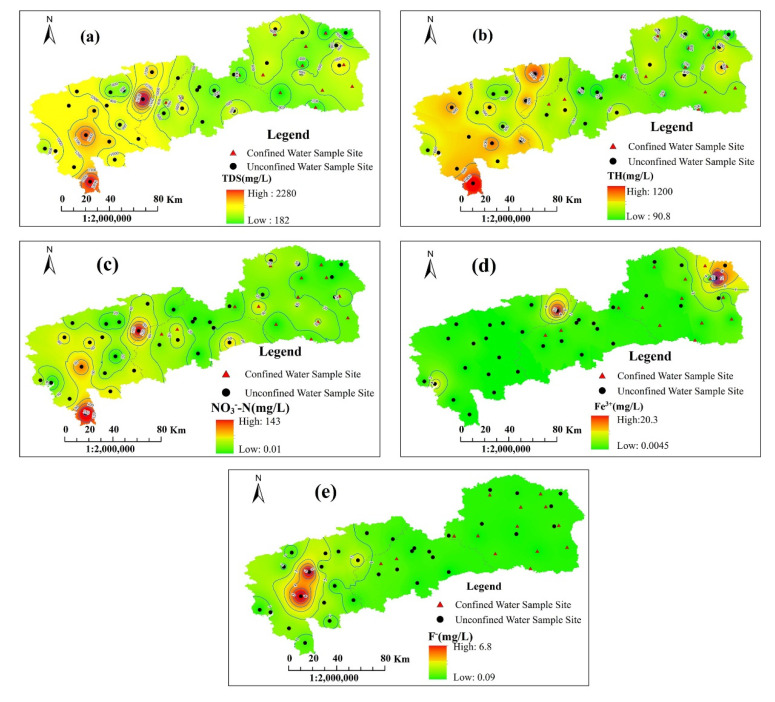
Spatial distribution of five indexes of groundwater in the study area: (**a**) TDS; (**b**) TH; (**c**) NO_3_^−^-N; (**d**) Fe^3+^; (**e**) F^−^.

**Figure 8 ijerph-19-09603-f008:**
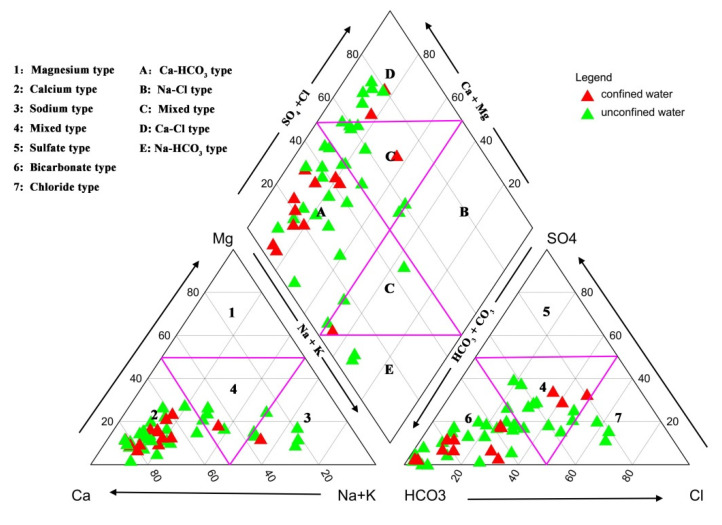
Piper diagram of 48 groundwater samples in the study area.

**Figure 9 ijerph-19-09603-f009:**
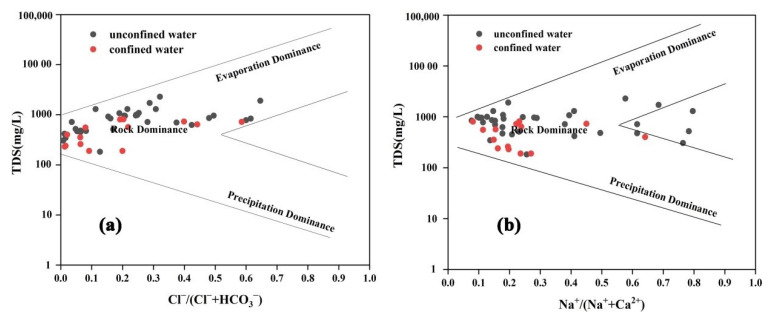
Gibbs diagrams of 48 groundwater samples in the study area. (**a**) TDS vs. [Cl^−^/(Cl^−^ + HCO_3_^−^)]. (**b**) TDS vs. [Na^+^/(Na^+^ + Ca^2+^)].

**Figure 10 ijerph-19-09603-f010:**
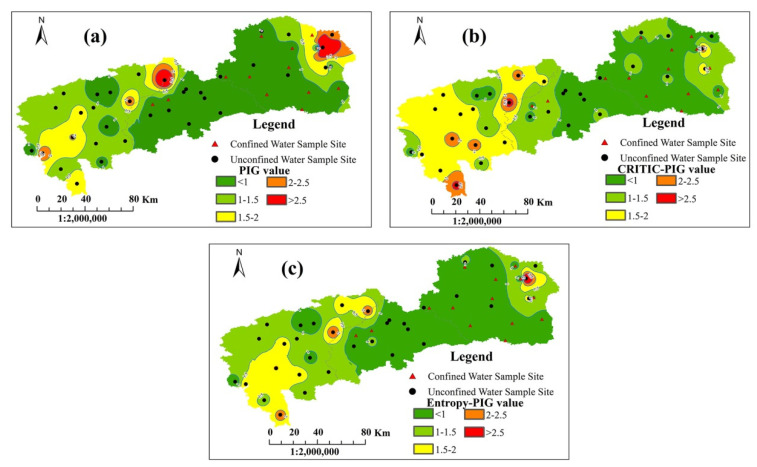
Distribution maps of PIG values, CRITIC-PIG values and Entropy-PIG values in the study area: (**a**) PIG values; (**b**) CRITIC-PIG values; (**c**) Entropy-PIG values.

**Table 1 ijerph-19-09603-t001:** Hydrogeological information of the aquifer systems of the study area.

Aquifer System	Aquifer	Lithology of the Aquifer	Permeability (m/d)	Thickness (m)	Water Inflow (m^3^/d)	Type of Groundwater
Quaternary Porous Aquifer System	Holocene Aquifer	Medium and coarse sand, gravel sand and gravel	30–100	5–20	500–3000	unconfined
	Upper Pleistocene (Guxiang Formation) Aquifer	Fine sand, sand and loss-shaped subclay	10–30	10–30	100–500	unconfined
	Middle Pleistocene (Huangshan Formation) Aquifer	Sand and loss-shaped subclay	Average	5–20	<100	unconfined
		Sand, gravel and clay	10–30	10–30	500–1000	confined
	Lower Pleistocene (Baitushan Formation) Aquifer	Sand, gravel and pebbles	Good	10–60	500–3000	unconfined
			Good	1–30	100–1000	confined
Cretaceous Fissure–Pore Aquifer System	Nenjiang Formation and Yaojia Formation Aquifer	Sandstone and mud rock	Bad	50–80	<100	confined
	Qingshankou Formation and Quantou Formation Aquifer	Sandstone, sandy conglomerate and mud rock	Bad	50–80	<100	

**Table 2 ijerph-19-09603-t002:** Values of allotted weight, weight parameter and drinking-water quality standard as well as the units of the 12 chemical parameters [15,20,21,37].

Chemical Parameters	Aw (Allotted Weight)	Wp (Weight Parameter)	Ds (Drinking-Water Quality Standard)	Unit
TDS	5	0.1136	500	mg/L
TH	4	0.0909	300	mg/L
Ca^2+^	2	0.0455	75	mg/L
Mg^2+^	2	0.0455	30	mg/L
Na^+^	4	0.0909	200	mg/L
K^+^	1	0.0227	12	mg/L
HCO_3_^−^	3	0.0682	300	mg/L
Cl^−^	4	0.0909	250	mg/L
SO_4_^2−^	5	0.1136	200	mg/L
NO_3_^−^	5	0.1136	45	mg/L
F^−^	5	0.1136	1.5	mg/L
Fe^3+^	4	0.0909	0.3	mg/L
Sum	44	1		

**Table 3 ijerph-19-09603-t003:** Five categories of water for drinking purposes according to three PIG values [15].

PIG	<1	1–1.5	1.5–2	2–2.5	>2.5
Result	Insignificant Pollution	Low Pollution	Moderate Pollution	High Pollution	Very High pollution

**Table 4 ijerph-19-09603-t004:** Statistical analysis results of physico-chemical parameters of unconfined water and confined water.

Parameter	Unit	Ds	Unconfined Water					Confined Water				
			Min	Max	Mean	S.D.	CV	Min	Max	Mean	S.D.	CV
pH	/	6.5–8.5	6.7	8.5	7.5	0.4	6	7	7.8	7.4	0.3	4
TDS	mg/L	500	182	2280	880.7	454	52	189	813	478.1	238	50
TH	mg/L	300	90.8	1200	555.8	269	48	120	522	297.9	137	46
Ca^2+^	mg/L	75	24.2	401	170.3	89	53	37.2	200	96.3	52	54
Mg^2+^	mg/L	30	3.9	84.8	28.1	22	77	4.25	24.9	12.3	7	60
Na^+^	mg/L	200	12.9	360	84.2	88	105	10.3	94.6	30.7	27	86
K^+^	mg/L	12	0.268	106	4.2	18	430	0.396	11.2	1.6	3	178
Cl^−^	mg/L	300	2.5	434	112.8	89	79	3.4	124	47.3	42	88
SO_4_^2−^	mg/L	250	1.55	298	102.3	86	84	5.77	163	42.8	46	108
HCO_3_^−^	mg/L	200	93	881	376.6	173	46	67.7	415	228.8	102	44
NO_3_^−^	mg/L	45	0.01	143	36.3	38	105	0.4	59.5	22.3	21	93
F^−^	mg/L	1.5	0.09	6.8	1.0	1.6	157	0.12	0.67	0.3	0.1	47
Fe^3+^	mg/L	0.3	0.0045	20.3	1.6	4.1	263	0.0045	4.97	0.5	1.3	247

S.D., standard deviation; CV, coefficient of variation; Ds, drinking-water quality standard.

**Table 5 ijerph-19-09603-t005:** Three PIG values and evaluation results of 34 unconfined groundwater samples.

Sample Number	PIG	Evaluation Result	CRITIC-PIG	Evaluation Result	Entropy-PIG	Evaluation Result
D1	0.670	Insignificant Pollution	1.213	Low Pollution	0.782	Insignificant Pollution
D2	2.552	Very High Pollution	1.778	Moderate Pollution	1.950	Moderate Pollution
D3	0.577	Insignificant Pollution	0.765	Insignificant Pollution	0.615	Insignificant Pollution
D4	4.109	Very High Pollution	1.635	Moderate Pollution	2.340	High Pollution
D5	0.891	Insignificant Pollution	1.329	Low Pollution	1.059	Low Pollution
D6	1.841	Moderate Pollution	2.584	Very High Pollution	2.091	High Pollution
D7	0.452	Insignificant Pollution	0.708	Insignificant Pollution	0.545	Insignificant Pollution
D8	1.033	Low Pollution	1.650	Moderate Pollution	1.281	Low Pollution
D9	1.453	Low Pollution	2.195	High Pollution	1.766	Moderate Pollution
D10	1.132	Low Pollution	1.615	Moderate Pollution	1.338	Low Pollution
D11	1.233	Low Pollution	1.746	Moderate Pollution	1.443	Low Pollution
D12	2.050	High Pollution	2.153	High Pollution	2.028	High Pollution
D13	1.229	Low Pollution	1.769	Moderate Pollution	1.446	Low Pollution
D14	1.906	Moderate Pollution	1.785	Moderate Pollution	1.703	Moderate Pollution
D15	1.033	Low Pollution	1.509	Moderate Pollution	1.226	Low Pollution
D16	0.593	Insignificant Pollution	0.881	Insignificant Pollution	0.688	Insignificant Pollution
D17	1.368	Low Pollution	2.138	High Pollution	1.637	Moderate Pollution
D18	2.282	High Pollution	2.795	Very High Pollution	2.447	High Pollution
D19	0.859	Insignificant Pollution	1.232	Low Pollution	0.983	Insignificant Pollution
D20	0.528	Insignificant Pollution	0.808	Insignificant Pollution	0.588	Insignificant Pollution
D21	1.037	Low Pollution	1.399	Low Pollution	1.161	Low Pollution
D22	7.114	Very High Pollution	2.568	Very High Pollution	3.985	Very High Pollution
D23	2.267	High Pollution	0.649	Insignificant Pollution	1.164	Low Pollution
D24	1.070	Low Pollution	1.036	Low Pollution	0.955	Insignificant Pollution
D25	0.856	Insignificant Pollution	1.204	Low Pollution	0.953	Insignificant Pollution
D26	1.931	Moderate Pollution	1.847	Moderate Pollution	1.681	Moderate Pollution
D27	0.927	Insignificant Pollution	1.371	Low Pollution	1.072	Low Pollution
D28	0.444	Insignificant Pollution	0.762	Insignificant Pollution	0.537	Insignificant Pollution
D29	0.570	Insignificant Pollution	0.884	Insignificant Pollution	0.679	Insignificant Pollution
D30	0.570	Insignificant Pollution	0.502	Insignificant Pollution	0.422	Insignificant Pollution
D31	0.561	Insignificant Pollution	0.776	Insignificant Pollution	0.524	Insignificant Pollution
D32	0.520	Insignificant Pollution	0.726	Insignificant Pollution	0.584	Insignificant Pollution
D33	0.447	Insignificant Pollution	0.747	Insignificant Pollution	0.507	Insignificant Pollution
D34	0.936	Insignificant Pollution	1.132	Low Pollution	0.987	Insignificant Pollution

**Table 6 ijerph-19-09603-t006:** Three PIG values and evaluation results of 14 confined groundwater samples.

Sample Number	PIG	Evaluation Result	CRITIC-PIG	Evaluation Result	Entropy-PIG	Evaluation Result
C1	0.204	Insignificant Pollution	0.294	Insignificant Pollution	0.229	Insignificant Pollution
C2	0.291	Insignificant Pollution	0.461	Insignificant Pollution	0.331	Insignificant Pollution
C3	0.282	Insignificant Pollution	0.443	Insignificant Pollution	0.326	Insignificant Pollution
C4	0.690	Insignificant Pollution	0.963	Insignificant Pollution	0.781	Insignificant Pollution
C5	0.630	Insignificant Pollution	0.850	Insignificant Pollution	0.716	Insignificant Pollution
C6	0.601	Insignificant Pollution	0.697	Insignificant Pollution	0.532	Insignificant Pollution
C7	0.819	Insignificant Pollution	1.190	Low Pollution	0.926	Insignificant Pollution
C8	0.552	Insignificant Pollution	0.853	Insignificant Pollution	0.626	Insignificant Pollution
C9	0.837	Insignificant Pollution	0.948	Insignificant Pollution	0.850	Insignificant Pollution
C10	0.350	Insignificant Pollution	0.472	Insignificant Pollution	0.365	Insignificant Pollution
C11	0.354	Insignificant Pollution	0.609	Insignificant Pollution	0.435	Insignificant Pollution
C12	1.697	Moderate Pollution	0.549	Insignificant Pollution	0.904	Insignificant Pollution
C13	0.622	Insignificant Pollution	0.925	Insignificant Pollution	0.718	Insignificant Pollution
C14	0.825	Insignificant Pollution	1.226	Low Pollution	0.959	Insignificant Pollution

**Table 7 ijerph-19-09603-t007:** Average values of the overall water quality of each chosen parameter in five pollution-level zones obtained using the PIG model.

TDS	TH	Ca^2+^	Mg^2+^	Na^+^	K^+^	HCO_3_^−^	Cl^−^	SO_4_^2−^	NO_3_^−^	F^−^	Fe^3+^	PIG	Pollution Level
0.126	0.105	0.067	0.022	0.020	0.002	0.066	0.021	0.024	0.058	0.033	0.057	0.602	Insignificant
0.237	0.228	0.147	0.054	0.034	0.002	0.098	0.051	0.077	0.118	0.060	0.071	1.177	Low
0.249	0.198	0.119	0.048	0.058	0.002	0.082	0.072	0.070	0.135	0.150	0.661	1.844	Moderate
0.317	0.170	0.085	0.077	0.098	0.068	0.082	0.053	0.105	0.202	0.243	0.699	2.200	High
0.199	0.193	0.110	0.050	0.027	0.004	0.092	0.044	0.111	0.001	0.041	3.720	4.592	Very High

**Table 8 ijerph-19-09603-t008:** Average values of the overall water quality of each chosen parameter in five pollution-level zones obtained using the CRITIC-PIG model.

TDS	TH	Ca^2+^	Mg^2+^	Na^+^	K^+^	HCO_3_^−^	Cl^−^	SO_4_^2−^	NO_3_^−^	F^−^	Fe^3+^	CRITIC-PIG	Pollution Level
0.211	0.164	0.080	0.008	0.015	0.002	0.157	0.010	0.012	0.013	0.004	0.033	0.708	Insignificant
0.405	0.330	0.182	0.012	0.016	0.001	0.174	0.033	0.024	0.037	0.003	0.015	1.233	Low
0.497	0.429	0.205	0.025	0.038	0.002	0.268	0.039	0.046	0.032	0.012	0.111	1.704	Moderate
0.708	0.550	0.248	0.036	0.066	0.002	0.356	0.048	0.070	0.050	0.027	0.002	2.162	High
0.843	0.582	0.290	0.025	0.054	0.054	0.224	0.073	0.083	0.082	0.010	0.329	2.649	Very High

**Table 9 ijerph-19-09603-t009:** Average values of the overall water quality of each chosen parameter in five pollution-level zones obtained using the Entropy-PIG model.

TDS	TH	Ca^2+^	Mg^2+^	Na^+^	K^+^	HCO_3_^−^	Cl^−^	SO_4_^2−^	NO_3_^−^	F^−^	Fe^3+^	Entropy-PIG	Pollution Level
0.084	0.124	0.142	0.050	0.018	0.003	0.078	0.013	0.028	0.044	0.018	0.055	0.657	Insignificant
0.142	0.246	0.287	0.100	0.021	0.002	0.101	0.026	0.072	0.099	0.030	0.115	1.243	Low
0.187	0.300	0.311	0.163	0.056	0.004	0.164	0.036	0.124	0.061	0.096	0.244	1.747	Moderate
0.264	0.306	0.313	0.184	0.081	0.068	0.109	0.058	0.132	0.204	0.116	0.392	2.227	High
0.153	0.299	0.319	0.068	0.010	0.006	0.114	0.024	0.206	0.000	0.014	2.771	3.985	Very High

## Data Availability

The datasets generated and/or analyzed during the current study are not publicly available.

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
