# Peer review of "Groundwater Quality Assessment in the Northern Part of Changchun City, Northeast China, Using PIG and Two Improved PIG Methods"

_ijerph, 2022, doi:10.3390/ijerph19159603_

Round 1

Reviewer 1 Report

The subject of your manuscript is appropriate for the journal. This manuscript is well structured and organized and presents a case study of interest to an international audience. Data is efficiently processed, well interpreted, and scientifically sound results. I would suggest this manuscript be reconsidered for publication after major revisions. A revised manuscript is required to emphasize the research problem and clarify puzzling issues related to materials and method and results and discussion sections. I encourage authors to consider the following remarks to strengthen their work.

Introduction

It is critical to distinguish what the present work is contributing to the overall body of literature. The introduction fails to set this context. It is important to clarify how this effort contributes to filling a defined knowledge gap. What is this study contributing that goes beyond what previous work has accomplished? Much of the effort of putting this work into a broader context is made at the introduction level. A good literature review is not measured in the number of citations as much as in drawing out the fundamental concepts the current work builds upon.

-It is necessary to add a paragraph (after the last one of the introduction section) that answers in the following questions:

(1)    What is the aim of your study?

(2)    What hypothesis is testing in your research paper?

(3)    how do you intend to answer the research question?

(4)    what is the originality and importance of your research?

Study area

It would be nice to use Köppen's climate classification.

Add more hydrogeological information such as hydraulic parameters and explain the permeability of each formation in detail.

This section needs an in-depth analysis of the geological setting.

It would be nice to have a geological or hydrogeological cross-section.

Materials

Why do you use dissolved concentrations for major ions? Do you filter the samples prior to analysis? If not, they would be total values and not the dissolved fraction. Were standard reference materials analyzed with samples? Were blanks and sample duplicates generated for QA/QC for each of the subsamples? Was a charge balance check done for water chemistry?

How did you determine TDS and TH? Did you calculate the previous-mentioned parameters?

Results and discussion

Table 3: Add the units as a new row in the table. In addition, add the detection and quantification limit of each parameter.

Table 3: Check all the manuscript for significant digits (e.g., replace 880.650 with 881, etc.)

 “As for Fe3+, a majority of areas are low in Fe3+ content, except for the northeast part of Nongan Country and Yushu City”: How do you explain these “extreme values”. As far as my knowledge, neutral to alkaline pH conditions do not favor Fe mobility. Please explain in detail this issue in the manuscript.

 “Gibbs diagrams are initially proposed by Gibbs [38] …”: According to the updated bibliography, the Gibbs plots are not useful and should be deleted (please study the following paper https://doi.org/10.1016/j.apgeochem.2018.07.009). This paper is based on Gibbs plot analysis, and it is very important to explain in the manuscript why, despite the doubts of Marandi and shad, you use the Gibbs diagrams. What makes them essential in this work? I have no problem using them, and to be honest, I agree with this approach, but you need to show that you are aware of this work and that you know what you did and why you did it.

 “Based on the background of the study area, anthropogenic sources are mainly the agricultural activities such as the excessive use of chemical fertilizers and pesticides, industrial activities and domestic waste.”: Explain that the intense use of agrochemical products and especially phosphate and nitrogen fertilizers, significantly degrade the groundwater quality and, according to cutting-edge research, are related to the occurrence of potentially toxic elements in groundwater (e.g., As, Cd, Cr, Cu, Zn); therefore affect human health (see 10.1007/s10661-019-7430-3).

Author Response

Please see the attachment,

Reviewer 2 Report

I believe that the article submitted to me for evaluation is a valuable scientific study. brings new information and modifies, and develops the research methods used so far on groundwater pollution. The work can be published with minor changes:

specific remarks:

verse 34: could you add more sources to this sentence and not be limited to only Chinese authors?

the disadvantage of the work is too few references to literature and the vast majority of authors from China are cited, could the authors make a broader literature review?

Fig 3 and Fig 5 would be more aesthetically pleasing if the captions were to fit in frames, I suggest reducing the font size

Round 2

Reviewer 1 Report

The manuscript was improved based on the comments! I suggest it for publication!